# Decline of Late Spring and Summer Snow Cover in the Scottish Highlands from 1984 to 2022: A Landsat Time Series

**Benedict D. Spracklen \* and Dominick V. Spracklen** 

School of Earth and Environment, University of Leeds, Leeds LS2 9JT, UK
* Correspondence: b10spracklen@gmail.com

**Abstract:** Late spring and summer snow cover, the remnants of winter and early spring snowfall, not only possess an intrinsic importance for montane flora and fauna, but also act as a sensitive indicator for climate change. The variability and potential trends in late spring and summer (snowmelt season) snow cover in mountain regions are often poorly documented. May to mid-September Landsat imagery from 1984 to 2022 was used to quantify changes in the snow-covered area of upland regions in the Scottish Highlands. There was substantial annual variability in the area of May to mid-September snow cover combined with a significant decline over the 39-year study period ($p = 0.02$). Long-term climate data used to show variability in May to mid-September snow cover was positively related to winter snowfall and negatively related to winter and April temperatures. The results from a long-running field survey counting the number of snow patches that survive until the following winter were used to check the veracity of the study. Further, accuracy was estimated through comparison with higher resolution Sentinel-2 imagery, giving a user and producer accuracy rate of 99.8% and 87%, respectively. Projected future warming will further diminish this scarce, valuable habitat, along with its associated plant communities, thus threatening the biodiversity and scenic value of the Scottish Highlands.

**Keywords:** satellite imagery; snow patches; climate change; remote sensing; threatened habitats; Scottish Highlands; seasonal snow cover

## 1. Introduction

Late spring and summer snow cover is an important component of montane ecosystems, strongly influencing the composition of vegetation cover [1–4]. Snow insulates vegetation from severe early frosts and damaging weather conditions [5–7] and provides protection against grazing herbivores [8]. In the late spring and summer (snowmelt season), melting snow cover provides an important source of water [9] and supports specialized plant communities (chinophytic vegetation) [10].

Perennial snow patches are a widespread feature of montane and arctic landscapes worldwide, and recent years have seen studies on this feature in Scotland [11], the Canadian arctic [12], Iceland [13], Australia [14] and Scandinavia [15]. Snow patch duration and survival is related to climatic factors such as winter snowfall and seasonal temperatures [11–15]. Declines in summer snow cover have been suggested in some regions such as the Snowy Mountains of Australia [14], but high interannual variability and issues with consistent monitoring can obscure potential trends. Long-term studies of snowmelt season snow cover are needed to help further identify potential trends.

The Scottish Highlands has seen an increase in mean minimum and maximum temperatures over the past century [16]. A 45-year-long study in the mountains of the Scottish Highlands linked a reduction in winter and spring snow cover duration over this period to an increase in mean temperature [17]. Climate models forecast a continued warming trend and further reductions in snow cover [18,19]. Small-scale semi-permanent snow patches and seasonal snow cover are potentially particularly vulnerable to this trend, lacking the

inherent buffering property of larger-scale glaciers and permanent snow. Consequently, summer snow cover can serve as an important indicator of climate change.

The Scottish Highlands, despite their relatively low elevation (<1350 m), contains seasonal, late-summer and semi-permanent snow cover. These have long been a source of interest and remark. For example, Samuel Johnson, on his famous visit with Boswell to Scotland, noted snow lying in late August 1773 on his passage through the West Highlands [20]. During the winter, the highest elevation areas will generally have extensive snow cover, though large-scale thaws are sometimes seen, and the snowline can fluctuate considerably [21]. By the start of May there is generally an ongoing rapid melt of the accumulated winter and early spring snowfall, and by July only a few small snow patches remain, with a tiny number of these sometimes surviving throughout the summer and into the autumn. These patches are remarkably faithful, occurring in the same spots year after year, generally where snow has been drifted by the prevailing south-westerly winds, and on north- and east-facing slopes that avoid the majority of sunlight.

Late summer snow patches in Scotland are associated with rare vegetation. For example, *Salix herbacea* is sometimes referred to as the snow willow due to its preference for growing in long-lasting snow patches, flowering as the last of the snow melts, whilst the frost-sensitive ferns *Cryptogramma crispa* and *Athyrium distentifolium* can cover large areas where they are protected from spring frost damage by long-lasting snow-lie. Especially late-lying areas of snow are important for specialised bryophyte communities, such as *Polytrichastrum sexangulare-Kiaeria starkei* [22], with the short growing season left after snowmelt insufficient for vascular vegetation to compete [10,23,24]. Many associated bryophyte species, such as *Conostomum tetragonum* and *Andreaea blyttii*, are of notable conservation value [25]. Furthermore, snow can serve as an important forage site for birds such as the rock ptarmigan (*Lagapus muta*) and snow bunting (*Plectrophenax nivalis*) [26], with insects trapped in the snow [27] or flushes of cranefly larvae hatching at the melting edge of the snow fields contributing an important food source during the breeding season [28].

Snow cover can be studied using ground surveys. For example, [11] surveyed July to October snow patches above 600 m in N.E. Scotland from 1974 to 1989 using visual observation from vantage points and counting the number and estimating the size of the espied areas. Since 1997, there have been annual reports of the number of snow patches surviving until the following winter across Scotland based on field surveys; see, for example, [29,30]. Field surveys were also used to map snow patches over a decade in the Snowy Mountains, Australia [14]. Field surveys and Google Earth imagery were used in a study in the Canadian Artic [12].

An alternative to field surveys is the utilization of optical satellite imagery, which has a long history of use in snow monitoring [31,32], with the MODIS [33–37] and Landsat [38–43] satellites frequently employed. This subject has recently been reviewed [44]. Landsat has sufficient spatial resolution (30 m) for use in the detection of even small snow patches and has an archive of uninterrupted imagery dating back to 1984. [13] used this Landsat archive to map perennial snow patches in northern Iceland, with the results validated by aerial photography and field surveys. [15] used Landsat imagery from 2000, 2004, 2006 and 2009 to map late summer snow patches in northern Finland. Further afield, Landsat tracked changes in snow cover extent from 1986 to 2018 in the South American Andes [45].

Despite decades of study, the variability and trends in summertime snow cover in the Scottish Highlands is still poorly quantified. The objective of this study is to use the almost four-decade-long Landsat image archive to quantify changes in the area of long-lying snow cover in the Scottish Highlands from 1984 to 2022. Scottish snow patches are the best-studied in the world, with a long-running ground survey tracking their survival. The results from Landsat can therefore be tested against this ground survey, and the method verified for use elsewhere. The study of snow in Scotland has overwhelmingly relied on field surveys, and to the best of our knowledge this paper is the first study to use medium-resolution satellite imagery to map Scottish snow patches and to track long-term variability in May to mid-September snow extent in Scotland.

## 2. Materials and Methods

### 2.1. Study Area

This study focused on nine high-altitude sites across the Scottish Highlands that are notable for the persistence of their snow cover in summertime: the Cairngorm Mountains, Lochnagar, Ben Alder, Creag Meagaidh, Ben Nevis, Glen Affric, Sgurr na Lapaich, the Fannichs and Black Mount (see Figure 1). These areas are characterized by open treeless landscapes of arctic-boreal calluna moor, acid grassland, Racomitrium heath, lochs and boulderfields [46–48]. The Köppen–Geiger climate classification for the study sites is largely KT (polar, tundra), with some small areas of lower elevation classified as Dfc (cold, no dry season, cold summer) [49]. The climate is strongly oceanic, with a high annual precipitation spread evenly through the year [50]. Due to the prevailing westerly winds, the western Atlantic Ocean-adjacent study sites have a higher mean annual precipitation than the eastern study sites. The easternmost two study sites of Cairngorms and Lochnagar both had an annual mean precipitation from 1991 to 2020 of less than 1900 mm/year (the mean for both study sites was 1864 mm/year), whilst the six more westerly study sites all had annual precipitations of greater than 2500 mm/year (the mean for these western study sites was 3040 mm/year). Temperatures were slightly higher for the western than the eastern subset, with mean annual temperatures from 1991 to 2020 of 3.44 °C and 3.78 °C.

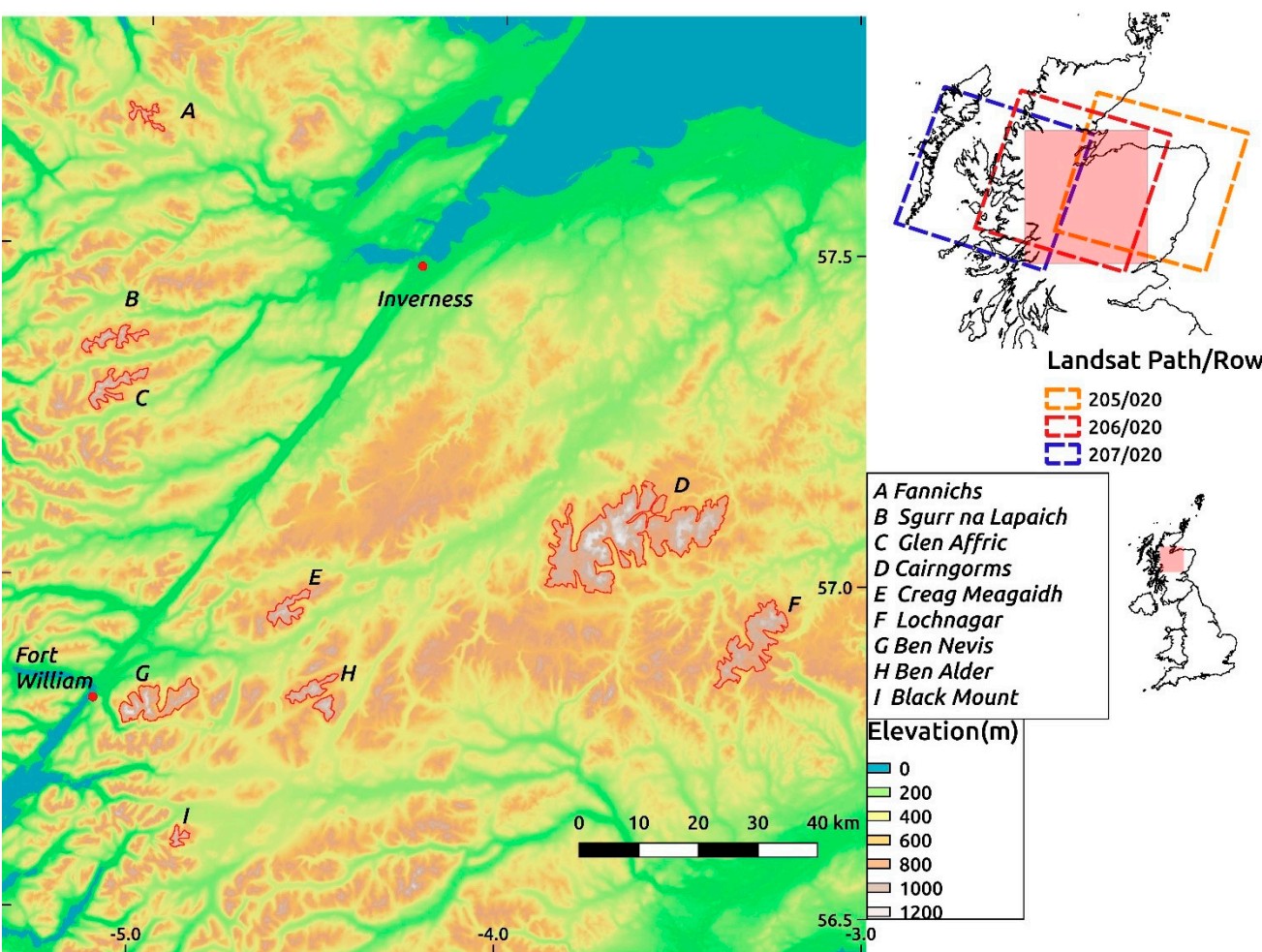

**Figure 1.** Map of location of study areas with background shading showing elevation (m).

In total, the study area covers 45,450 ha, of which the Cairngorms comprises about half. All nine study areas lie within about 1° latitude and 2° longitude of each other and are of similar elevation, ranging from a minimum of about 800 m to a maximum of 1345 m.

### 2.2. Landsat Imagery

Scotland is of sufficiently northern latitude to be covered twice over by Landsat: all sites are covered by Path/Row 206/020; the eastern four sites of Cairngorm, Lochnagar, Creag Meagaidh and Ben Alder by 205/020 and the remainder by 207/020 (see Figure 1) All Landsat 5, 7, 8 and 9 imagery from these Path/Rows for 1st May to mid-September from 1984 to 2022 (inclusive) was downloaded from https://earthexplorer.usgs.gov (accessed on 15 September 2022).

Cloud, cloud shadow and Landsat malfunction pixels were removed from the study by setting them to nodata. For a given Landsat image, if a site was covered by more than 60% valid pixels, then it was included in the study. In total, 222 images were used over the 39-year study period (see Figure 2).

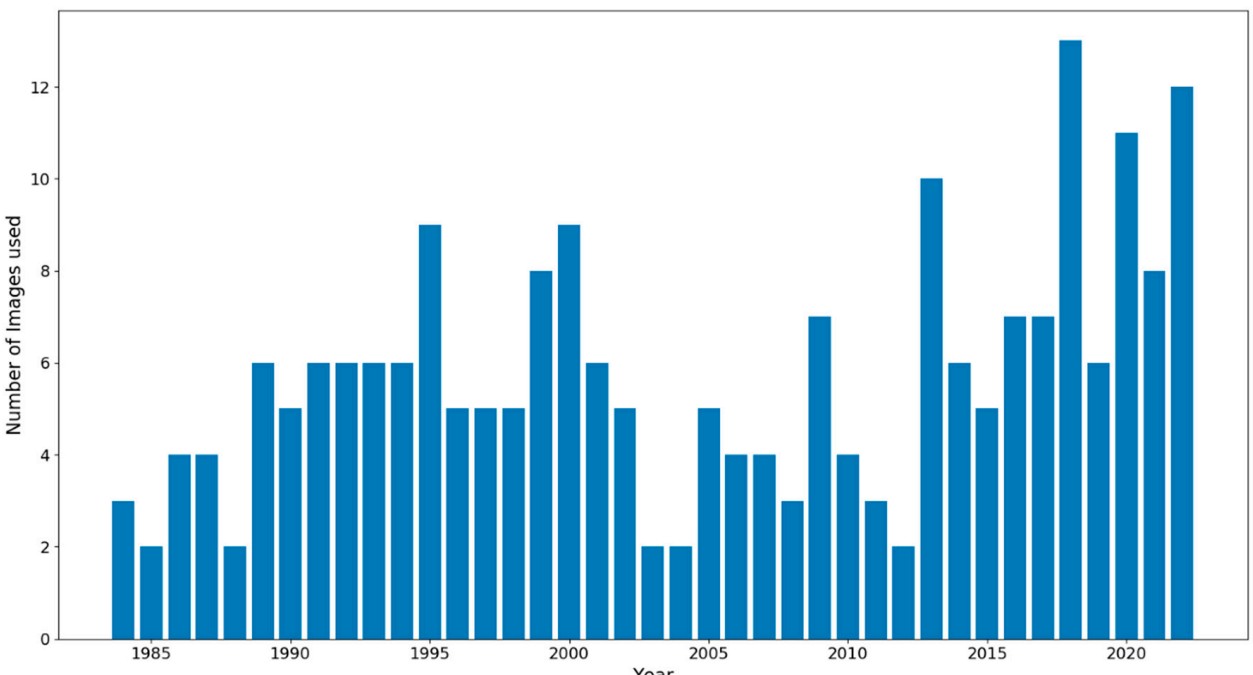

**Figure 2.** Number of Landsat images used to determine snow cover for each year in the study.

A small number of the utilised Landsat 5 images were significantly misaligned, sometimes by as much as 20 km. Three such images were used from 1994, one from 1998 and one from 2009. These five images were correctly realigned using Ground Control Points (GCPs), locations whose latitude and longitude were known precisely. At least seven GCPs were used for each misaligned image.

### 2.3. Processing of Landsat Imagery

2.3.1. Computation of NDSI

The normalised snow difference index (NDSI) uses the fact that snow reflects strongly in the visual wavelengths and absorbs strongly over the shortwave infrared (SWIR) wavelengths, helping to distinguish snow from other visually reflective features such as clouds or rocks. This index, or close variants thereof, is very widely used to detect snow cover in satellite imagery [41,51–57] and was computed as:

$$NDSI = \frac{\rho_{green} - \rho_{SWIR}}{\rho_{green} + \rho_{SWIR}} \tag{1}$$

where ρ is the reflectance (see Figure 3).

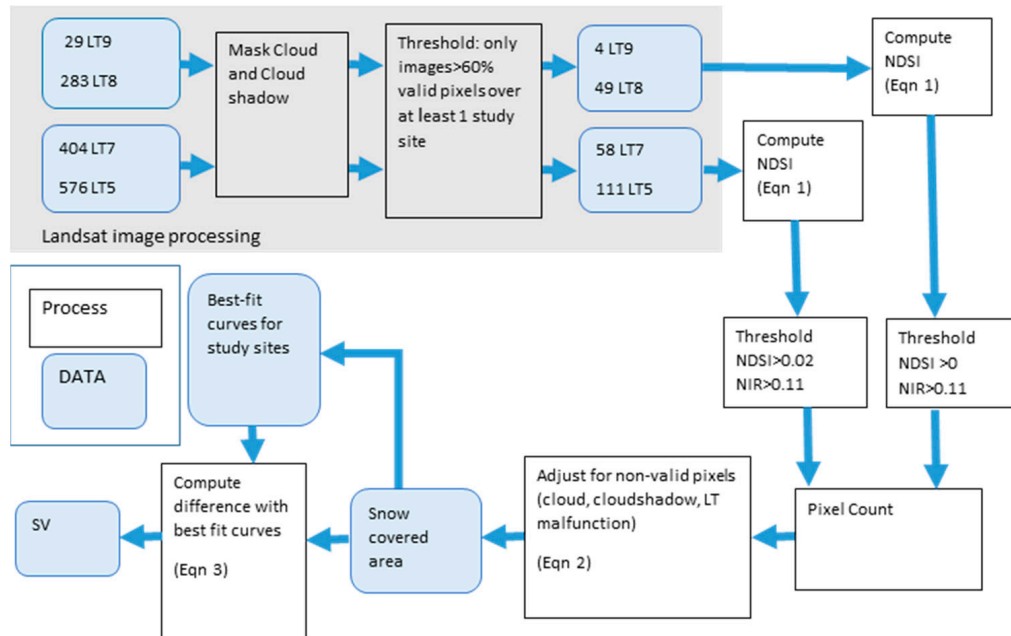

**Figure 3.** Overall workflow to determine snow cover value (SV) from Landsat imagery. LT is the abbreviation for Landsat; Eqn for Equation.

### 2.3.2. Calibration and Thresholds

The wavelengths used in Landsat 8–9 are slightly different from those used in Landsat 5–7 (see Table 1). Landsat 7 was launched in 1999, Landsat 8 in 2013 and Landsat 9 in 2021, whilst Landsat 5 ceased operation in 2011. Therefore, there was a switch from using entirely Landsat 5–7 imagery pre-2013, to using a mix of Landsat 7 and Landsat 8 and 9 from 2013 onwards, with 32 of the 85 post-2012 images being Landsat 7. Mismatch in band-derived indices between Landsat 7 and 8 has been noted previously [58]. It was noticed that Landsat 5–7 seems to give higher NDSI values for snow cover than Landsat 8–9, presumably due to the shift in wavelengths used. Therefore, Landsat 5–7 gives higher snow cover values than Landsat 8–9. There was fortunately an effective way to quantify this difference—on several dates there was a Landsat 7 image followed or preceded just a day before or after by a Landsat 8 image. For example, a cloud-free Landsat 8 image on 25 June 2018 was followed by a cloud-free Landsat 7 image on 26 June 2018. Since it is unlikely that snow cover shifted appreciably during the course of just one day, it is possible to see how much the two satellites differ in their estimation of snow cover and calibrate them so that they give the same number of snow-covered pixels. Landsat 8 imagery from 25 June 2018, 26 June and 28 June 2019 and Landsat 7 imagery from 24 June 2018 and 27 June 2019 was used for this purpose. It was found that a Landsat 8 threshold value of 0 corresponded to a Landsat 7 threshold value of 0.02. Accordingly, the NDSI threshold was set as 0 for Landsat 8 and 9, and 0.02 for Landsat 5 and 7. The choice of 0 was dictated by a rapid rise in noise (pixels falsely identified as snow that contained no snow) below this value. Shadowed areas and the boundaries of waterbodies were most prone to noise pixels.

To help eliminate water-covered pixels, which can have NDSI values in the same range as snow, the method used by [59] in applying a further condition that the NIR band reflectance be greater than 0.11 was followed. This condition applied to all the Landsat imagery. To summarize, any valid pixel with both an NDSI value above 0 (Landsat 8 and 9) or 0.02 (Landsat 5, 7) and an NIR value above 0.11 was qualified as snow-covered.

**Table 1.** Wavelengths of Landsat satellites. All the featured bands have a resolution of 30 m.

| Bands | Landsat 5/7 Wavelength (μm) | Landsat 8/9 Wavelength (μm) |
|---|---|---|
| Blue | 0.45–0.52 | 0.45–0.51 |
| Green | 0.52–0.6 | 0.53–0.59 |
| Red | 0.63–0.69 | 0.64–0.67 |
| Near-Infrared (NIR) | 0.76–0.9 | 0.85–0.88 |
| Infrared (SWIR) | 1.55–1.75 | 1.57–1.65 |
| Infrared (SWIR) | 2.08–2.35 | 2.11–2.29 |

The number of snow-covered pixels lying within each study site was counted for every image in the study, giving the snow-covered area. Cloud, cloud shadow and Landsat malfunction meant that many of our images had variable areas of non-valid pixels, where any snow would not be counted. To make the results from these different images more comparable to each other, we assumed that in any given image, the fractional snow-covered area under non-valid pixels was the same as that under the valid pixels. Therefore, snow-covered area was adjusted by being multiplied by the fraction of study site area and valid pixel area as follows:

$$snow\text{-}covered\ area = snow\text{-}covered\ area_{unadjusted} \times \frac{Studysite\ pixel\ number}{Studysite\ valid\ pixel\ number} \quad (2)$$

2.3.3. Computation of Snow Cover Values

For each of the nine sites, all the results from 1984 to 2022 for a particular site were used to plot a best-fit snowmelt curve, plotting the snow-covered area for that site on the y-axis against the numerical day of the year that particular snow cover value had been recorded. A negative exponential best-fit curve was used for all sites. This produces nine snow-melt curves, one for each site, giving a "best-fit" value for snow-covered area for every day of the study season. The difference in the best-fit snow-covered area and the actual snow-covered area was used to indicate how snowy the site is compared to how snowy it usually has been over the study period. More specifically, the difference between the base-10 logarithm of the best-fit area and the base-10 logarithm of the actual snow-covered area was defined to be the "snow cover value" ($SV_i$) for that particular date and site:

$$SV_i = \log_{10}(snow\text{-}covered\ area) - \log_{10}(best\text{-}fit\ snow\text{-}covered\ area) \quad (3)$$

The reason for using logarithms is to avoid a situation where results early in the season, when difference values are likely to be larger, overshadow results from later in the season, when less snow remains and differences are likely to be smaller. In this way, positive $SV_i$ indicates more snow cover than normal, and a negative $SV_i$ indicates less snow than normal. For all the results from all the sites in a year, the average was taken to give a single mean snow cover value (SV) for that year, producing a time series of SV values for each year.

The Mann–Kendall test [60,61] and the Theil–Sen estimator [62,63] were used to quantify trend significance and trend magnitude respectively in the mean snow cover value (SV) time series. The Mann–Kendall test is a rank-based non-parametric test of whether a significant monotonic trend exists in a time series. It is not affected by missing values or outliers and produces a z-value which can be used to calculate the significance level (*p*-value). The Theil–Sen estimator is a robust non-parametric method of estimating the magnitude of a linear trend which is relatively insensitive to outliers. Both the Mann–Kendall test and the Theil–Sen estimator are sensitive to serial correlation in a time series. [64] This occurs where the residuals are not independent of each other, or, in other words, when the past values can influence later values. Except for a tiny number of miniscule snow patches,

all Scottish snow melts before the arrival of fresh snow of the new winter season, and in this respect, SV should not be correlated. The annual values of SV are therefore regarded as independent.

Trend significance and magnitude were computed for SV for (1) all nine study sites together; (2) an eastern subset (Cairngorms and Lochnagar) and (3) a western subset of study sites (Black Mount, Ben Nevis, Glen Affric, Sgurr na Lapaich, Creag Meagaidh, Ben Alder and the Fannichs). All subsequent references to SV are for Case (1) all study sites, unless specifically noted otherwise.

All the above image processing and statistical analysis was carried out using Python 2.7.

### 2.4. Accuracy Assessment

Accuracy assessment for snow cover was done using Sentinel-2 (S-2) satellite imagery, which has a higher 10–20 m resolution. The first S-2 satellite was launched in 2015, and coverage continues uninterrupted to the present day. Sentinel-2 images were downloaded that either coincided with the day of a Landsat image or were within a single day of their corresponding Landsat image. Snow cover on the S-2 image was then mapped using the RGB image and compared to the Landsat mapped results.

Further, the results from a long-running ground survey of the number of snow patches (SN) that survive melting in the summer and autumn months and persist into the next season's fresh snowfall was used. This survey gives an SN number for each year from 1997 to 2022, ranging from 0 (no surviving snow patches) to 74 (in 2015). The correlation of this series with the SV values for the same years was calculated. Initially both series were detrended by taking the first differences:

$$SV' = SV_t - SV_{t-1} \tag{4}$$

$$SN' = SN_t - SN_{t-1} \tag{5}$$

where subscript *t* indicates value at year *t*, so that subscript *t*−1 indicates the value from the previous year. The Kendall tau correlation coefficient was then computed on the resulting SV′ and SN′.

### 2.5. Climate Data

The HadUK-Grid climate data was obtained from the Met Office (https://www.metoffice.gov.uk/research/climate/maps-and-data/data/haduk-grid/datasets, accessed 15 January 2023). This dataset has a 1 km × 1 km grid resolution covering all of the UK and all of the study period save the final year, 2022. The variables used were principally precipitation and temperature data and can be seen in Table 2 below. They were calculated as the mean value over all the study sites combined.

Temperature and precipitation were largely agglomerated into summer and winter categories, with winter and summer consisting of November to March and May to September, respectively. April is generally the principal snowmelt month in Scotland, and so was considered separately and in more detail. It was assumed that all precipitation in December, January and February was snowfall. For November, March and April, daily precipitation and maximum air temperature data was used to estimate snowfall by qualifying any precipitation on a day with a temperature of below 3 °C as snowfall.

Variables were checked for multicollinearity using the Variance Inflation Factor (VIF). Any variables with a VIF of over five were removed. Multiple linear regression using the remaining variables, with mean SV value as the dependent variable, was then computed.

**Table 2.** Summary of climate variables used in the study. Nov., Dec., Jan., Feb., Aug. and Sep. are November, December, January, February, August and September, respectively. For winter snowfall for the starred months of November and December, daily temperature and precipitation data were used to compute snowfall.

| Variable | Data Used | Months | Description | Units |
|---|---|---|---|---|
| Winter temperature | tas (monthly) | Nov., Dec., Jan., Feb., March | Summed mean daily air temperature | Celsius |
| Winter snowfall | rainfall (monthly) tasmax (daily) * rainfall (daily) * | Nov. *, Dec., Jan., Feb., March * | Summed total snowfall | mm |
| Winter wind | sfcWind (monthly) | Nov., Dec., Jan., Feb., March | Summed mean wind speed at 10 m above ground | knots |
| Summer temperature | tas (monthly) | May, June, July, Aug., Sep. | Summed mean daily air temperature | Celsius |
| Summer precipitation | rainfall (monthly) | May, June, July, Aug., Sep. | Summed total precipitation | mm |
| Summer sunshine | sun (monthly) | May, June, July, Aug., Sep. | Summed duration of bright sunshine | hours |
| April temperature | tas (monthly) | April | Mean daily air temperature | Celsius |
| April thaw days | tasmax (daily) | April | Number of days with max temperature over 3 Celsius | Days |
| April snowfall | tasmax (daily) rainfall (daily) | April | Total snowfall | Celsius |

* indicates months for which daily data was used to compute snowfall.

## 3. Results

### 3.1. Site Snowmelt Curves

Linear and quadratic curves were tried, but a negative exponential decay curve proved to be the best fit, with $R^2 > 0.7$ for all nine sites (see Figure 4).

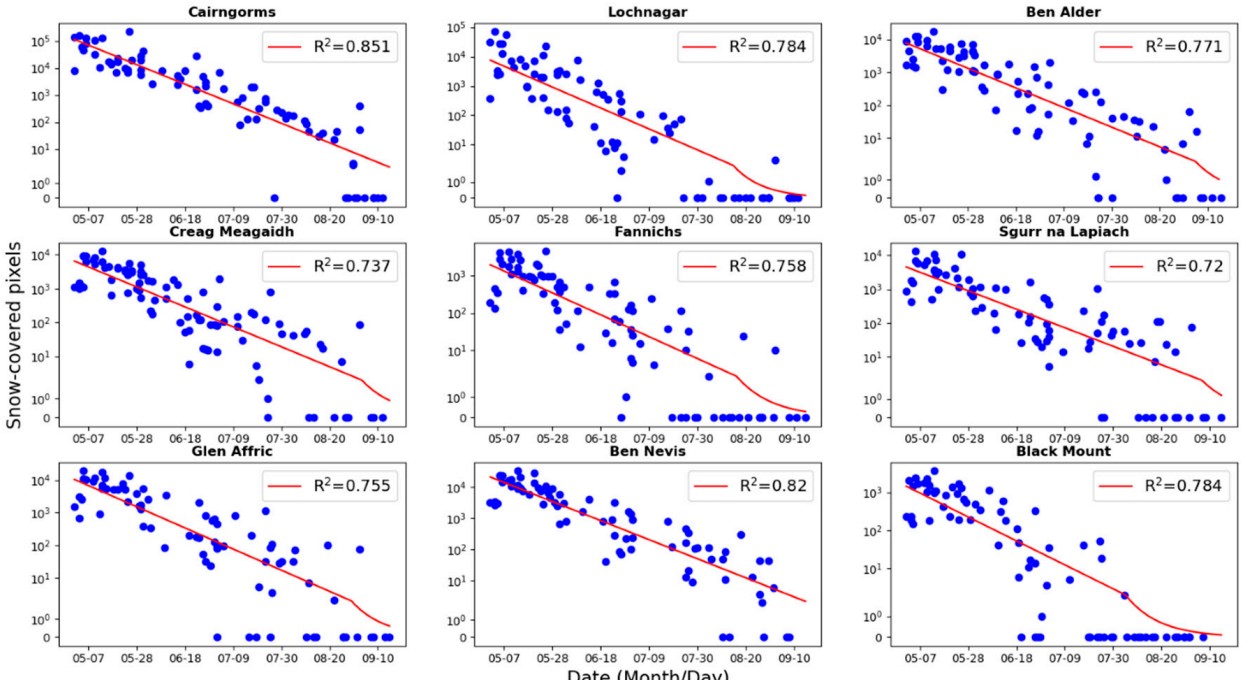

**Figure 4.** Curves of snow-covered pixels versus date for the nine sites in the study area. The red line shows the best-fit line for each site. Note the logarithmic scale of the y-axis.

*3.2. Accuracy Estimates and Validation Tests*

3.2.1. Accuracy Estimate Using Sentinel-2 Imagery

The ground-truthing was carried out through comparison of selected Landsat imagery with higher-resolution Sentinel-2 imagery. The confusion matrix for 25 June 2018 for Landsat 8 can be seen below (Table 3). The other confusion matrices can be found in the supplementary tables (Supplementary Tables S1–S9).

**Table 3.** Confusion matrix showing accuracy of Landsat 8 25 June 2018 satellite image in classifying snow-covered pixels, using 25 June 2018 Sentinel-2 image for validation. All numbers are pixel counts. User accuracy is 100% and producer accuracy is 76.5%.

| | | **Landsat** | | |
|---|---|---|---|---|
| | | Snow | No snow | Total |
| Sentinel-2 | Snow | 552 | 170 | 722 |
| | No snow | 0 | 394,146 | 394,146 |
| | Total | 552 | 394,316 | 394,868 |

Snow is an extremely distinctive feature of upland, treeless terrain and the user accuracy was consequently very high (see Table 4). User accuracy is the snow-covered area that Landsat correctly classified divided by the area that Landsat classified as snow covered. The producer accuracy was notably lower, due to the number of small snow patches that lay beneath the spatial resolution of Landsat imagery. Producer accuracy is the snow-covered area that Landsat correctly classified divided by the snow-covered area given by Sentinel-2. Producer accuracy was quite variable, ranging from just over 50% to over 90%, influenced by whether the snow lay in small patches or in larger areas. Producer accuracy for the Nevis range was poorer than for the other study sites because the Landsat satellite had limited success in picking out even larger areas of snow situated in the shadow of the large north-facing cliffs that lie adjacent to the Ben Nevis summit, whilst the Sentinel-2 satellite saw this snow clearly. Overall accuracy for all ten accuracy verification studies was 99.9%.

**Table 4.** Details of all the images used in accuracy verification. Sentinel and Landsat date are the date of the Sentinel and Landsat images used, respectively (month/day). Valid pixels indicates the number of valid Landsat pixels overlaying the study sites that are covered by both the specified Landsat and Sentinel image. User and Prod. indicate User and Producer accuracy for that image, respectively.

| Number | Landsat 7/8 | Year | Sentinel Date | Sentinel Granule | Landsat Date | Landsat Path/Row | Valid Pixels | User (%) | Prod.(%) |
|---|---|---|---|---|---|---|---|---|---|
| 1 | 8 | 2016 | 06/02 | VVU | 06/01 | 207/020 | 95,916 | 99.8 | 93.2 |
| 2 | 8 | 2016 | 06/02 | VVJ | 06/03 | 205/020 | 44,712 | 100 | 91.2 |
| 3 | 8 | 2018 | 06/25 | VVJ | 06/25 | 205/020 | 394,868 | 100 | 76.5 |
| 4 | 8 | 2019 | 06/27 | VVU | 06/26 | 207/020 | 113,848 | 97.9 | 63.6 |
| 5 | 8 | 2019 | 06/27 | VVJ | 06/28 | 205/020 | 394,871 | 99.8 | 75.0 |
| 6 | 8 | 2021 | 07/01 | VVU | 07/01 | 207/020 | 49,729 | 99.5 | 78.2 |
| 7 | 8 | 2022 | 06/04 | VVJ | 06/04 | 205/020 | 384,916 | 99.5 | 84.7 |
| 8 | 7 | 2018 | 06/25 | VVJ | 06/24 | 206/020 | 290,618 | 100 | 67.2 |
| 9 | 7 | 2019 | 06/27 | VVJ | 06/27 | 206/020 | 281,764 | 99.5 | 64.4 |
| 10 | 7 | 2021 | 07/01 | VVU | 07/02 | 206/020 | 28,902 | 100 | 53.5 |
| Total | — | — | — | — | — | — | 2,080,144 | 99.8 | 87.0 |

In 2003, the Landsat 7 satellite suffered a malfunction in the Scan Line Corrector, which meant that subsequent imagery is crossed by evenly spaced lines of nodata pixels, their breadth widening with distance from the centre of the image. Snow lying under these nodata lines account for the inferior producer accuracy of Landsat 7 compared to 8. For the Landsat 7 image areas with valid pixels, producer accuracy was the same as for Landsat 8.

### 3.2.2. Comparison with Snow Patch Snow Survey

After taking first differences to remove any possible linear trend, a Kendall–Tau coefficient of 0.41 ($p$ = 0.004) was found, indicating a statistically significant correlation between the annual snow cover values (SV) and the number of remaining snow patches.

### 3.3. Snow Cover Time Series

### 3.3.1. Decline in Snow Cover from 1984 to 2022

Snow cover was found to be highly variable, with considerable annual variation (see Figure 5). The snowiest year was 1994, and the least snowy was 2003. For all study sites, the Mann–Kendall Test found a significant decline in SV from 1984 to 2022 ($z$ = −2.44, $p$ = 0.015). The Theil–Sen Regression showed an annual decline of −0.02 (95% C.I. of −0.03 and 0). Over the whole study period, the average SV value was 0, with a standard deviation of 0.53. Four years (1986, 1988, 1994 and 2015) had an SV value above a single standard deviation from the study period mean, with 1998, 2003, 2005, 2007, 2011, 2017, 2019 and 2022 having an SV below one standard deviation.

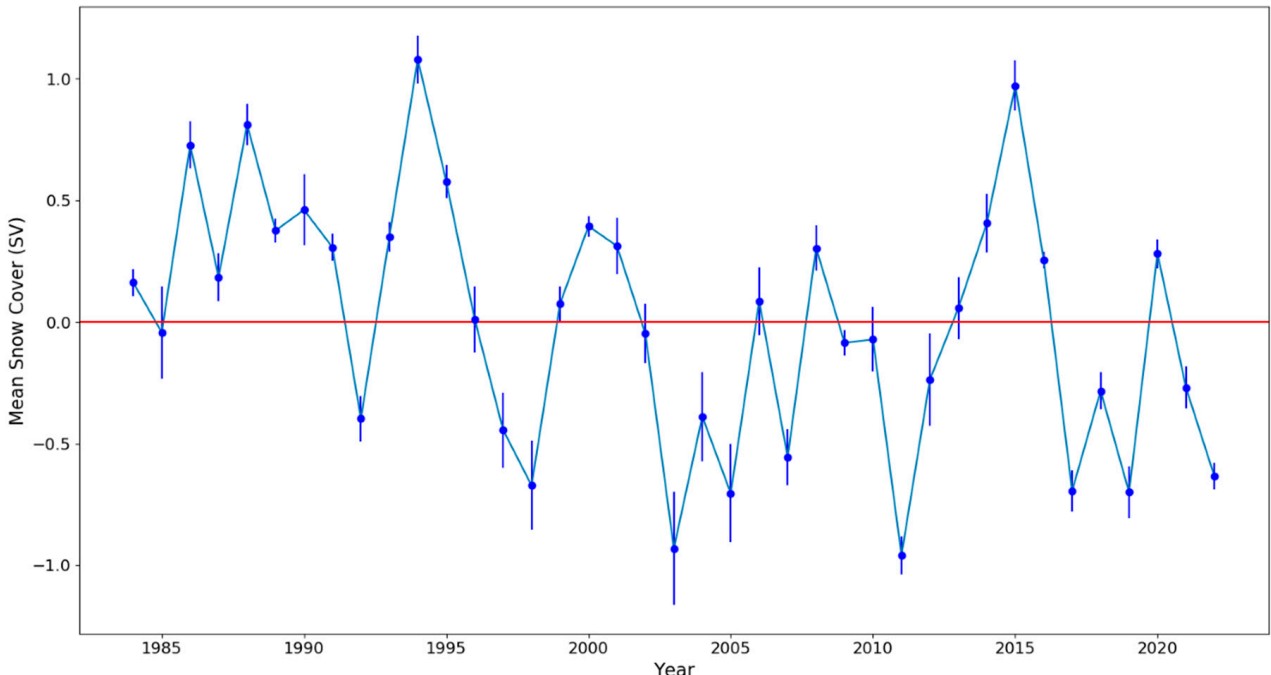

**Figure 5.** Mean annual snow cover from 1984 to 2022. Error bars show standard error. Positive values of SV indicate more snow than normal, negative values indicate less than normal.

For the eastern set of study sites (Cairngorms and Lochnagar), there were 103 Landsat images covering the study period. The Mann–Kendall test still found a significant decline in SV from 1984 to 2022 ($z$ = −2.24, $p$ = 0.025). For the western subset of study sites (Black Mount, Ben Nevis, Glen Affric, Sgurr na Lapaich, Creag Meagaidh, Ben Alder and the Fannichs), there were 189 Landsat images, from which a significant decline in SV was also found ($z$ = −2.18, $p$ = 0.03).

3.3.2. Impact of Cloud Cover and Landsat Malfunction

Cloud, cloud shadow and Landsat 7 malfunction meant that 60% of the study site Landsat images used had at least some nodata pixels. A total of 22% of study site images had at least 15% non-valid pixels and 38% had at least 5% non-valid pixels. The mean percentage of invalid pixels for all of the study sites was 8%. If we removed from our study all the study site images with at least 5% non-valid pixels, reducing the number of Landsat images used from 222 to 158, our findings of a decline in snow cover are not significantly altered, with the Mann–Kendall test continuing to give a significant decline in SV over the study period (z = −2.49, $p$ = 0.01).

Of the 58 Landsat 7 images in our study, the 41 post-2002 images suffered from the Scan Line Corrector malfunction, which results in all but the centre of the imagery being crossed by lines of non-valid pixels. Removing these 41 Landsat 7 images resulted in a total of 181 Landsat images covering the study period, but did not impact the results, with the Mann–Kendall test continuing to find significant decline in SV over the study period ($p$ = 0.015, z = −2.44).

*3.4. Impact of Climatic Variables*

A linear regression model of nine climate variables was used to model SV. Multicollinearity (indicated by VIF values exceeding five) resulted in the removal of the April thaw days, April snowfall, winter wind, summer rain and summer sunshine variables. The model was left with four variables, of which three were significant ($p$ < 0.05): winter snowfall, winter temperature and April temperature (see Table 5).

**Table 5.** Significant independent variables in linear regression model predicting mean snow cover value. Model $R^2$ = 0.58. C.I.s indicate Confidence Intervals.

| Variable | Coefficient | 97.5% C.I.s: Lower Bound | 97.5% C.I.s: Upper Bound | *p*-Value |
|---|---|---|---|---|
| Winter temperature | −0.06 | −0.09 | −0.03 | <0.001 |
| Winter snowfall | 0.0016 | 0.001 | 0.002 | <0.001 |
| April temperature | −0.14 | −0.23 | −0.05 | 0.003 |

Of the nine climate variables considered, the Mann–Kendall and Theil–Sen regression tests showed a significant increase ($p$ = 0.005, z = 2.8) in the summed summer temperature of 0.1 °C per year over the study period (95% Confidence Intervals between 0 and 0.2 °C). There was also a significant decline ($p$ = 0.04, z = −2) in the summed winter wind speed of –0.3 knots (95% Confidence Intervals of –0.5 and 0 knots). Warming winter (0.08 °C per year, $p$ = 0.2, z = 1.3) and April (0.03 °C per year, $p$ = 0.25, z = 1.1) temperatures were observed consistent with a decline in snow cover, but the trends were not significant.

**4. Discussion**

Landsat satellite imagery was used to explore variability and potential trends in May to mid-September snow cover in the Scottish Highlands over the period of 1984–2022. High cloud cover which obscures the surface was the biggest hindrance to the study. Cloud-free imagery from 222 dates was used, though these were not evenly spread over the 39-year study period. There were just two images from 1985, 2003, 2004 and 2012. Satellite coverage improved in 2013, and again in 2022, with the launch of Landsat 8 and 9, respectively. The possibility of using Synthetic Aperture Radar (SAR) imagery to monitor snow cover could be investigated [65], such as the publicly available Sentinel-1 imagery, with a spatial resolution of 20 m, and over Scotland, a temporal resolution of a few days. SAR imagery is largely immune to cloud and weather conditions. However, there is no substitute for the lengthy Landsat archive.

In the study period, snow in the Scottish Highlands completely melted in the years 1996, 2003, 2006, 2017 and 2022 [66–69]. The results from a field survey enumerating the surviving

snow patches aligned with the SV values with a high correlation. The years with the lowest SV values were 2003, 2005, 2011, 2017 and 2019, which had 0, 2, 2, 0 and 1 surviving snow patches, respectively. From 1997 to 2022, the year with the highest number of remaining snow patches was 2015, with 74, which also possessed the highest SV for that period.

A significant decline in May to mid-September snow cover in the Scottish Highlands (Mann–Kendall Test; z = −2.44, p = 0.015) over the period of 1984–2022 was found. A field study in the eastern Scottish Highlands from 1974 to 1989 [11] that counted the number and estimated the area of summer snow patches found a decline in both number and size that was not significant. A MODIS satellite study of year-round snow cover extent in Iceland [37] from 2000 to 2018 found a significant increase in snow cover for the month of June. A field study in the Australian Snowy Mountains [14] claimed a significant decline in summer snow patches. Both of these study sites have maritime climates like that experienced in Scotland, with the Snowy Mountains being particularly similar, as only a few small snow patches, if any, survive a typical austral summer. A Landsat study in the South American Andes [45] found significant declines in austral summer snow cover extent over the study period of 1986–2018. There have been numerous MODIS satellite studies in the High Mountain Area of Asia, these showing strong spatial and temporal heterogeneity in changes in snow cover extent [70–73].

In a field study of the Scottish Highlands [11], temperature in winter and spring and snow-drift in spring were the most important factors influencing summer snow cover. This significant effect of winter and spring temperatures was also found in this study's analysis of climatic variables. A significant effect for winter snowfall was found, which was not indicated in the previous analysis. A field survey in the Snowy Mountains of Australia [14] found that winter snowfall, wind direction and summer warmth all influenced snow patch duration. An Icelandic satellite survey [13] also showed winter snowfall and mean annual temperature to be important factors.

Earlier melting of summer snow is likely to impact specialised plant communities. The British National Vegetation Classification [22] notes three of these specialised communities of snow-associated montane vegetation: the northern haircap *Polytrichum sexangulare* and Starke's fork moss *Kiaeria starkei* (NVC U11), dwarf willow *Salix herbacea* and bristly fringe moss *Racomitrium heterostichum* (NVC U12) and the parsley fern *Cryptogramma crispa* and alpine lady fern *Athyrium distentifolium* (NVC U18). This comprises a seventh of the acid grassland and montane communities identified. Further work [10] has split off the additional bryophyte dominated communities: *Marsupella brevisima -Anthelia juratzkana* and *Pohlia ludwigii*. Further underlying the importance of these habitats, 27 bryophyte species associated with snow cover are qualified as either Nationally rare or Nationally Scarce [74], including 3 IUCN Red List Endangered and 6 Red List Vulnerable species, and the strictly protected (Wildlife and Countryside Act-Schedule 8) listed liverwort pointed frostwort (*Gymnomitrion apiculatum*) [75,76]. The diminution of late snow cover therefore significantly threatens the biological diversity of the Scottish Highlands, as these specialised communities are replaced by more competitive species [77].

An obvious follow-on study would be to determine if this study's recorded decline applies across other coastal-adjacent sites, such as Scandinavia. In the future, the introduction of Sentinel-2 satellites in 2015 and the continuation of the Landsat series should allow for uninterrupted further monitoring of snow cover.

## 5. Conclusions

Despite the small size and number of the Scottish Highland's long-lying snow cover, they hold considerable conservation and scenic importance and provide a sensitive indicator of climate change. Using Landsat imagery from 1984 to 2022, a considerable annual variation in the area of May to mid-September (late spring and summer) snow cover was found. May to mid-September snow cover was positively related to winter snowfall and negatively related to winter and April temperatures. Summer snow cover has been more extensively studied in Asia, the Americas and continental Europe than in the coastal fringe

of Europe, with studies often finding marked spatial variation in response to climatic change, thus highlighting the need for further detailed regional studies in this area. This study found a significant decline in May to mid-September snow cover over the study period, underlining the threat of climate change and posing a risk to montane biodiversity and ecosystem services in the Scottish Highlands.

**Supplementary Materials:** The following supporting information can be downloaded at: https://www.mdpi.com/article/10.3390/rs15071944/s1. These materials supply the confusion matrices for the accuracy assessments described in Section 2.4. Table S1: Confusion matrix showing accuracy of Landsat 8 1 June 2016 satellite image in classifying snow-covered pixels, using June 2nd 2016 Sentinel-2 image for validation; Table S2: Confusion matrix showing accuracy of Landsat 8 3 June 2016 satellite image in classifying snow-covered pixels, using 2 June 2016 Sentinel-2 image for validation; Table S3: Confusion matrix showing accuracy of Landsat 8 26 June 2019 satellite image in classifying snow-covered pixels, using 27 June 2019 Sentinel-2 image for validation; Table S4: Confusion matrix showing accuracy of Landsat 8 28 June 2019 satellite image in classifying snow-covered pixels, using 27 June 2019 Sentinel-2 image for validation; Table S5: Confusion matrix showing accuracy of Landsat 8 1 July 2021 satellite image in classifying snow-covered pixels, using 1 July 2021 Sentinel-2 image for validation; Table S6: Confusion matrix showing accuracy of Landsat 8 4 June 2022 satellite image in classifying snow-covered pixels, using 4 June 2022 Sentinel-2 image for validation; Table S7: Confusion matrix showing accuracy of Landsat 7 24 June 2018 satellite image in classifying snow-covered pixels, using 25 June 2018 Sentinel-2 image for validation; Table S8: Confusion matrix showing accuracy of Landsat 7 27 June 2019 satellite image in classifying snow-covered pixels, using 27 June 2019 Sentinel-2 image for validation; Table S9: Confusion matrix showing accuracy of Landsat 7 2 July 2021 satellite image in classifying snow-covered pixels, using 1 July 2021 Sentinel-2 image for validation.

**Author Contributions:** Conceptualization, B.D.S.; Methodology, B.D.S.; Software, B.D.S.; Validation, B.D.S.; Formal Analysis, B.D.S.; Investigation, B.D.S.; Resources, D.V.S.; Data Curation, B.D.S.; Writing—Original Draft Preparation, B.D.S. and D.V.S.; Writing—Review and Editing, B.D.S. and D.V.S.; Visualization, B.D.S.; Supervision, D.V.S.; Project Administration, D.V.S.; Funding Acquisition, D.V.S. All authors have read and agreed to the published version of the manuscript.

**Funding:** The research has been supported by funding from the European Research Council (ERC) under the European Union's Horizon 2020 research and innovation programme (DECAF project, Grant Agreement No. 771492) and the Natural Environment Research Council (NE/L013347/1).

**Data Availability Statement:** Publicly available datasets were analyzed in this study. Landsat data can be found here: https://earthexplorer.usgs.gov/ (accessed on 15 September 2022). Climate data can be found here: https://www.metoffice.gov.uk/research/climate/maps-and-data/data/haduk-grid/datasets (accessed on 15 January 2023).

**Conflicts of Interest:** The authors declare no conflict of interest.

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
