# Peer review of "Decline of Late Spring and Summer Snow Cover in the Scottish Highlands from 1984 to 2022: A Landsat Time Series"

_remotesensing, doi:10.3390/rs15071944_

Round 1
Reviewer 1 Report
The study seems promising and contains a good methodology to study the long term changes in snow patches of late spring and summer. However, there are some major flaws in the study regarding selection of data and ignoring the area under clouds and shadow. Also authors should mention the limitations of their research.
Comments
Authors can provide a methodology flowchart for better understanding of the research methodology.
There are several limitations to the Thiel-Sen estimator. Are authors sure that these limitations will not interfere in the study?
(Line 130) If a site was covered by more than 60% valid pixels then it was included in the study.
Neglecting nearly 40% of the area that falls under shadow and clouds is huge; most of the areas under shadow in the mountains contain snow/ice that is preserved due to lower temperature in shadow zone. If these areas are nearly 30-40% of the study area then excluding nearly one third of the area can exclude permanent snow/ice available in the study area.
(Line 299-301) In 2003 the Landsat-7 satellite suffered a malfunction in the Scan-Line Corrector, which meant that subsequent imagery is crossed by evenly spaced lines of nodata pixels, their breadth widening with distance from the centre of the image
When correct data of landsat 5 is available till 2013 and landsat 8 is available 2013 onwards, why use landsat 7? Despite knowing a large part of the images contain no data values. These no data values that falls over the actual area with snow wouldn’t calculate the snow area in years after 2003 thus it will automatically reduce the total area of snow in those images post 2003, which is a major flaw.
(Line 352-353) The years with the lowest SV values were 2003, 2005, 2011, 2017 and 2019 which had 0, 2, 2, 0 and 1 surviving snow patches respectively.
Results also snow that post 2003 lowest SV values were detected which can be influenced by a large part of snow covered area not being included due to no data values in landsat 7. (Authors can check whether no data values in images lie over actual snow in the ground by manually comparing the locations of no data value with Google earth images or some other data such as landsat 5)
· Using landsat 7 data is a major flaw of the study, using data that doesn’t contain correct representation of actual ground features can highly influence overall results. Excluding 40% of the area from images because of cloud cover and shadow is also not recommended in studies like this. It’s better to use only clean data that can be used for accurate mapping of snow.
Author Response
We thank the reviewer for these helpful comments, and for raising an important issue in our study that we had not sufficiently addressed. We respond to all the Reviewer comments in turn. Reviewer comments are in bold type, our responses below.
The study seems promising and contains a good methodology to study the long term changes in snow patches of late spring and summer. However, there are some major flaws in the study regarding selection of data and ignoring the area under clouds and shadow. Also authors should mention the limitations of their research.
Comments
Authors can provide a methodology flowchart for better understanding of the research methodology.
RESPONSE: Thanks for this suggestion. New Figure 2 added.
There are several limitations to the Thiel-Sen estimator. Are authors sure that these limitations will not interfere in the study?
RESPONSE: We add the following to Section 2.3.3 (Ln206): “Both the Mann-Kendall test and the Thiel-Sen estimator are sensitive to serial correlation in timeseries. [64] This occurs where the residuals are not independent of each other, or in other words, when the past values can influence later values. Except for a tiny number of miniscule snow patches, all Scottish snow melts before the arrival of fresh snow of the new winter season, and in this respect SV should not be correlated. The annual values of SV are therefore regarded as independent “ We change Lines 203-204 to “The Mann-Kendall test is a rank-based non-parametric test of whether a significant monotonic trend exists in a time-series.”
Added references:
Blain, G. C. (2014). Removing the influence of the serial correlation on the Mann-Kendall test. Revista Brasileira de Meteorologia, 29, 161-170.
(Line 130) If a site was covered by more than 60% valid pixels then it was included in the study.
Neglecting nearly 40% of the area that falls under shadow and clouds is huge; most of the areas under shadow in the mountains contain snow/ice that is preserved due to lower temperature in shadow zone. If these areas are nearly 30-40% of the study area then excluding nearly one third of the area can exclude permanent snow/ice available in the study area.
RESPONSE: Thanks for these important comments. We had not well-explained our approach. We now provide a more detailed explanation of our method and we add analysis (Section 3.3.2, see response to comment below) to explore the impacts of the issues raised by the Reviewer.
Permanent shadow caused by cliffs etc was not an issue in our study areas. Only cloud, cloud-shadow and Landsat malfunction caused non-valid pixels. Of course, cloud is always an issue with optical satellite imagery. The 60% threshold applies to study sites in individual Landsat images, so the overall amount of the study area across multiple images is much higher than this. We change Line 125 to “Cloud, cloud-shadow and Landsat malfunction pixels were removed from the study by setting to nodata. For a given Landsat image, if a site was covered by more than 60% valid pixels then it was included in our study.”
(Line 299-301) In 2003 the Landsat-7 satellite suffered a malfunction in the Scan-Line Corrector, which meant that subsequent imagery is crossed by evenly spaced lines of nodata pixels, their breadth widening with distance from the centre of the image
When correct data of landsat 5 is available till 2013 and landsat 8 is available 2013 onwards, why use landsat 7? Despite knowing a large part of the images contain no data values. These no data values that falls over the actual area with snow wouldn’t calculate the snow area in years after 2003 thus it will automatically reduce the total area of snow in those images post 2003, which is a major flaw.
RESPONSE: We thank the Reviewer for bringing these issues to our attention. Firstly, we left out an important step in our Materials and Method section. For images with nodata pixels, either through cloud, cloud-shadow or Landsat malfunction (all treated the same), we multiplied up the number of snow covered pixels in proportion to the area that was not valid. We add the following to the Materials and Methods Section (line 178):
“The number of snow covered pixels lying within each study site was counted for every image in the study, giving snow covered area. Cloud, cloud-shadow and Landsat malfunction meant that many of our images had variable areas of non-valid pixels, where any snow would not be counted. To make the results from these different images more comparable to each other, we assumed that the fractional snow covered area under non-valid pixels was the same as that under the valid pixels. Therefore, snow covered area was multiplied by the fraction of study site area and valid pixel area as follows:
EQUATION”
This assumes that fractional snow-covered area is the same for valid and non-valid pixels. We examine the validity of this assumption by adding a further analysis and subsection to the Results section: “Impact of cloud cover and Landsat malfunction” Here we investigate if excluding the study sites with more than 5% non-valid pixels changes our findings and add the following subsection:
“3.3.2 Impact of Cloud Cover and Landsat malfunction
Cloud, cloud-shadow and Landsat-7 malfunction meant that 60% of the study site images used had at least some nodata pixels. 22% of study site images had at least 15% non-valid pixels and 38% at least 5% non-valid pixels. The mean percentage of invalid pixels for all the study sites was 8%. If we removed from our study all the study site images with at least 5% non-valid pixels, reducing the number of Landsat images used from 222 to 158, our findings of a decline in snow cover are not significantly altered, with the Mann-Kendall test continuing to give a significant decline in SV over the study period (z=-2.49, p=0.01.)
(Line 352-353) The years with the lowest SV values were 2003, 2005, 2011, 2017 and 2019 which had 0, 2, 2, 0 and 1 surviving snow patches respectively.
Results also snow that post 2003 lowest SV values were detected which can be influenced by a large part of snow covered area not being included due to no data values in landsat 7. (Authors can check whether no data values in images lie over actual snow in the ground by manually comparing the locations of no data value with Google earth images or some other data such as landsat 5)
Using landsat 7 data is a major flaw of the study, using data that doesn’t contain correct representation of actual ground features can highly influence overall results. Excluding 40% of the area from images because of cloud cover and shadow is also not recommended in studies like this. It’s better to use only clean data that can be used for accurate mapping of snow.
RESPONSE: Thanks for this comment. We had not fully explained our approach or provided sufficient information on how cloud, cloud shadow and Landsat 7 data impact our analysis. We have now provided additional details which we hope better explains our approach.
We use the images with the lowest amount of cloud cover that are available. Unfortunately the frequent cloud cover over Scotland meant that over half of our study site images had at least some nodata pixels. The study would not be feasible using only non-cloud covered imagery due to the low fraction of images that meet this criteria. We note that we do not exclude 40% of the study area, but that 60% of the images contain at least some non-valid pixels (we have now clarified this in Section 3.2.2). The No-data lines on Landsat-7 imagery doesn't cover all the imagery – the central area of the image has full coverage- for example Ben Nevis study site for Landsat Path/Row 207/020 – is not effected by the Scan-Line Corrector malfunction, and in clear conditions has 100% valid pixel coverage.
We test the impacts of including Landsat-7 in our analysis. If we remove the 41 post Scan Line Corrector malfunction Landsat-7 images (out of 58 total Landsat-7 images used in the study) from our analysis, leaving us with 181 Landsat images covering our study period, it makes no difference to our results, with the Mann-Kendall test still giving a significant decline (p=0.015, z=-2.44.) We add the following text (section 3.2.2):
“Of the 58 Landsat-7 images in our study, the 41 post-2002 images suffered from the Scan Line Corrector malfunction, which results in all but the centre of the imagery being crossed by lines of non-valid pixels. Removing these 41 Landsat-7 images, resulted in a total of 181 Landsat images covering the study period, but did not impact the results, with the Mann-Kendall test continuing to find significant decline in SV over the study period (p=0.015, z=-2.44).”
We hope that our fuller explanation of our method, and our addition of the section 3.3.2 to the Results quantifying the impact of non-valid pixels on our results helps to deal with this issue. This additional analysis demonstrates that we get similar results using our reduced subsection of imagery to exclude images with >5% non-valid pixel coverage, and that the trends we identify are not sensitive to the number of images we use or the thresholds we apply. For this reason we would like to continue to utilise as many study site images as possible, as we feel that more dates and more coverage for each year helps improve the robustness of our results, even if some of the imagery has areas of non-valid data.
Reviewer 2 Report
The manuscript titled “Decline of late spring and summer snow patches in the Scottish Highlands from 1984-2022: a Landsat timeseries” by Spracklen and Spracklen is within the scope of the journal. The study utilises Landsat satellite imagery from 1984-2022 to quantify changes in the snow-covered area for upland regions of the study area (Scottish Highlands). The study is very important as it assess the changes in the snow cover which are essential part of the habitat in the study area thus responsible for the biodiversity.
Introduction portion can be strengthened more by citing more references.
Authors should try to avoid the pronouns “we” and “our”.
For assessing accuracy of classification why not overall accuracy was considered?
There are a few Grammatical and formatting errors in the manuscript. For eg:-
· Line 10-11 “We use May to mid-September Landsat imagery from 1984-2022 to quantify changes in the snow-covered area for upland regions of the Scottish Highlands.” Use should be replaced with have used.
The discussion part can be enhanced.
“A negative exponential decay curve proved to be an excellent fit,” What other functions authors experimented with, of which the results were less fitting?
Author Response
We thank the reviewer for these helpful comments. Reviewer comments are in bold type, our responses below.
The manuscript titled “Decline of late spring and summer snow patches in the Scottish Highlands from 1984-2022: a Landsat timeseries” by Spracklen and Spracklen is within the scope of the journal. The study utilises Landsat satellite imagery from 1984-2022 to quantify changes in the snow-covered area for upland regions of the study area (Scottish Highlands). The study is very important as it assess the changes in the snow cover which are essential part of the habitat in the study area thus responsible for the biodiversity.
Introduction portion can be strengthened more by citing more references.
RESPONSE: Changed to: “An alternative to field surveys is the use of optical satellite imagery, which has a long history of use in snow monitoring [31,32], with the MODIS[33-37] and Landsat[38-43] satellites frequently used. This subject has recently been reviewed [44].”
and:
“Further afield, Landsat tracked changes in snow cover extent from 1986-2018 in the South American Andes [45].”
Added references are: Meier, M. F. (1975). Application of remote-sensing techniques to the study of seasonal snow cover. Journal of Glaciology, 15(73), 251-265.
Schneider, S., & Matson, M. (1977). Satellite observations of snowcover in the Sierra Nevadas during the great California drought. Remote Sensing of Environment, 6(4), 327-334.
Lopez, P., Sirguey, P., Arnaud, Y., Pouyaud, B., & Chevallier, P. (2008). Snow cover monitoring in the Northern Patagonia Icefield using MODIS satellite images (2000–2006). Global and Planetary Change, 61(3-4), 103-116.
Marchane, A., Jarlan, L., Hanich, L., Boudhar, A., Gascoin, S., Tavernier, A., ... & Berjamy, B. (2015). Assessment of daily MODIS snow cover products to monitor snow cover dynamics over the Moroccan Atlas mountain range. Remote Sensing of Environment, 160, 72-86.
Deng, G., Tang, Z., Hu, G., Wang, J., Sang, G., & Li, J. (2021). Spatiotemporal dynamics of snowline altitude and their responses to climate change in the Tienshan Mountains, Central Asia, During 2001–2019. Sustainability, 13(7), 3992.
Tang, Z., Wang, X., Deng, G., Wang, X., Jiang, Z., & Sang, G. (2020). Spatiotemporal variation of snowline altitude at the end of melting season across High Mountain Asia, using MODIS snow cover product. Advances in Space Research, 66(11), 2629-2645.
Gunnarsson, A., Garðarsson, S. M., & Sveinsson, Ó. G. (2019). Icelandic snow cover characteristics derived from a gap-filled MODIS daily snow cover product. Hydrology and Earth System Sciences, 23(7), 3021-3036.
Park, S. H., Lee, M. J., & Jung, H. S. (2016). Spatiotemporal analysis of snow cover variations at Mt. Kilimanjaro using multi-temporal Landsat images during 27 years. Journal of Atmospheric and Solar-Terrestrial Physics, 143, 37-46.
McFadden, E. M., Ramage, J., & Rodbell, D. T. (2011). Landsat TM and ETM+ derived snowline altitudes in the Cordillera Huayhuash and Cordillera Raura, Peru, 1986–2005. The Cryosphere, 5(2), 419-430.
Cordero, R. R., Asencio, V., Feron, S., Damiani, A., Llanillo, P. J., Sepulveda, E., ... & Casassa, G. (2019). Dry-season snow cover losses in the Andes (18–40 S) driven by changes in large-scale climate modes. Scientific Reports, 9(1), 16945.
Authors should try to avoid the pronouns “we” and “our”.
RESPONSE: Changed this throughout the paper: for example Ln 12 “We use May to mid-September Landsat...” changed to “May to mid-September Landsat imagery was used...”
Similarly for example Ln15 “significant decline over our 39-year study...” changed to “ significant decline over the 39-year study...”
For assessing accuracy of classification why not overall accuracy was considered?
RESPONSE: Because of the preponderance of non-snow covered pixels, we thought the overall accuracy was not that informative, with 99.9% overall accuracy for the ten accuracy verification studies, and all individual overall accuracies greater than 99.4%. We add to Section 3.2.1 (Ln 295): “Overall accuracy for all ten accuracy verification studies was 99.9%”
There are a few Grammatical and formatting errors in the manuscript. For eg:-
Line 10-11 “We use May to mid-September Landsat imagery from 1984-2022 to quantify changes in the snow-covered area for upland regions of the Scottish Highlands.” Use should be replaced with have used.
RESPONSE: Changed as suggested. We switch to past tense throughout the script
The discussion part can be enhanced.
RESPONSE: We expand the discussion by increasing comparisons to previous literature:
“A MODIS satellite study of year-round snow cover extent in Iceland [37] from 2000-2018 found a significant increase in snow cover for the month of June. A field study in the Australian Snowy Mountains [14] claimed a significant decline in summer snow patches. Both these study sites have maritime climates like that experienced in Scotland, with the Snowy Mountains being particularly similar as only a few small snow patches, if any, survive a typical austral summer. A Landsat study in the South American Andes [45] found significant declines in austral summer snow cover extent over the study period of 1986 to 2018. There have been numerous MODIS satellite studies in the High Mountain Area of Asia, these showing strong spatial and temporal heterogeneity in changes in snow cover extent [70-73]. “
“A negative exponential decay curve proved to be an excellent fit,” What other functions authors experimented with, of which the results were less fitting?
RESPONSE: We tried quadratic and linear fits to the data. Changed to: “Linear and quadratic curves were tried, but a negative exponential decay curve proved to be the best fit, with R2>0.7 for all nine sites (see Figure 3.)”
Reviewer 3 Report
The work is interesting. Below are some suggestions for improving the manuscript.
The authors used the pronoun "We" more than 20 times in the manuscript. Please revise the MS and avoid the use of "We." Also in some places in the manuscript you are using “Our”. The use of the possessive pronoun is not preferred in writing research papers. Please correct these errors throughout the manuscript.
Abstract and introduction: the novelty of the study is unclear. What is the novelty of this study? Please clarify the study's objective and novelty.
Study area: Please include a map of the study area in the United Kingdom, while keeping the current map.
Equations: each equation should have a number.
Discussion: I propose that the authors widen their discussion by comparing their findings to those of previous studies in the literature.
Conclusions: The conclusion should be written and extended to include the recommendations as well as the significance of this study for decision makers.
References: Please use the MDPI reference style.
Author Response
We thank the reviewer for these helpful comments. Reviewer comments are in bold type, our responses below.
The work is interesting. Below are some suggestions for improving the manuscript.
The authors used the pronoun "We" more than 20 times in the manuscript. Please revise the MS and avoid the use of "We." Also in some places in the manuscript you are using “Our”. The use of the possessive pronoun is not preferred in writing research papers. Please correct these errors throughout the manuscript.
RESPONSE: Changed this throughout the paper: for example Ln 12 “We use May to mid-September Landsat...” changed to “May to mid-September Landsat imagery was used...”
Similarly, for example, Ln15 “significant decline over our 39-year study...” changed to “ significant decline over the 39-year study...”
Abstract and introduction: the novelty of the study is unclear. What is the novelty of this study? Please clarify the study's objective and novelty.
RESPONSE: The Introduction changed to: “Study of snow in Scotland has overwhelmingly relied on field surveys, and to the best of our knowledge this paper is the first study to use medium-resolution satellite imagery to map Scottish snow patches and to track longterm variability in May to mid-September snow extent in Scotland.”
Study area: Please include a map of the study area in the United Kingdom, while keeping the current map.
RESPONSE: Figure 1 adjusted as suggested.
Equations: each equation should have a number.
RESPONSE: Equations modified as suggested.
Discussion: I propose that the authors widen their discussion by comparing their findings to those of previous studies in the literature.
RESPONSE: We expand the discussion by increasing comparisons to previous literature:
“A MODIS satellite study of year-round snow cover extent in Iceland [37] from 2000-2018 found a significant increase in snow cover for the month of June. A field study in the Australian Snowy Mountains [14] claimed a significant decline in summer snow patches. Both these study sites have maritime climates like that experienced in Scotland, with the Snowy Mountains being particularly similar as only a few small snow patches, if any, survive a typical austral summer. A Landsat study in the South American Andes [45] found significant declines in austral summer snow cover extent over the study period of 1986 to 2018. There have been numerous MODIS satellite studies in the High Mountain Area of Asia, these showing strong spatial and temporal heterogeneity in changes in snow cover extent [70-73]. “
Conclusions: The conclusion should be written and extended to include the recommendations as well as the significance of this study for decision makers.
RESPONSE: We add to the Conclusion: “Summer snow cover has been more extensively studied in Asia, the Americas and continental Europe than in the coastal fringe of Europe, with studies often finding marked spatial variation in response to climatic change, thus highlighting the need for further detailed regional studies in this area.”
References: Please use the MDPI reference style.
RESPONSE: Altered to this style: “15. Kivinen, S.; Kaarlejärvi, E.; Jylhä, K.; Räisänen, J. Spatiotemporal distribution of threatened high-latitude snowbed and snow patch habitats in warming climate. Environmental Research Letters 2012, 7(3), 034024. “
Reviewer 4 Report
Using landsat data, this paper analyzes the snow cover changes during snowmelt period of Scottish Highlands from 1984 to 2022, and draws the conclusion that snow cover is decreasing. The data used are reliable and the conclusion is reasonable. However, the presentation of some important results needs to be improved. The expression of some professional words is not standard enough.
Special Comments:
1. The ‘late spring and summer’changed to ‘snowmelt period’ or ‘snowmelt season’. ‘snow patches’ changed to ‘snow cover’.
2. Line 45/52, change semi-permanent snow cover to seasonal snow cover
3. in the 1. Introduction and 4. Discussion, the existing studies on monitoring snow cover changes by remote sensing should be added, and relevant comparison and discussion should be increased.
4. Figure 4 is an important result supporting the conclusion of this paper, but it is insufficient to reflect the characteristics of the spatial distribution of remote sensing data in this paper with only a linear wave chart of interannual changes. It is suggested to increase the spatial expression.
Suggested references:
[1] Satellite observed spatiotemporal variability of snow cover and snow phenology over High Mountain Asia from 2002 to 2021. Journal of Hydrology,2022, 613, 128438.
[2] Spatiotemporal dynamics of snowline altitude and their responses to climate change in the Tienshan Mountains, Central Asia, During 2001–2019 [J]. Sustainability, 2021, 13(7): 3992.
[3] Spatiotemporal variation of snowline altitude at the end of melting season across High Mountain Asia, using MODIS snow cover product [J]. Advances in Space Research, 2020, 66(11): 2629-2645.
[4] Spatiotemporal variation of snow cover in Tianshan Mountains, Central Asia, based on cloud-free MODIS fractional snow cover product, 2001–2015 [J]. Remote Sensing, 2017, 9(10): 1045.
Author Response
We thank the reviewer for these helpful comments. Reviewer comments are in bold type, our responses below.
Using Landsat data, this paper analyzes the snow cover changes during snowmelt period of Scottish Highlands from 1984 to 2022, and draws the conclusion that snow cover is decreasing. The data used are reliable and the conclusion is reasonable. However, the presentation of some important results needs to be improved. The expression of some professional words is not standard enough.
Special Comments:
1. The ‘late spring and summer’changed to ‘snowmelt period’ or ‘snowmelt season’. ‘snow patches’ changed to ‘snow cover’.
RESPONSE: 'Late spring and summer' changed to 'snowmelt season' or 'May to mid-September' .Changed 'snow patches' to 'snow cover' where appropriate, such as in title.
2. Line 45/52, change semi-permanent snow cover to seasonal snow cover
RESPONSE: added “seasonal snow cover” to Lines
3. in the 1. Introduction and 4. Discussion, the existing studies on monitoring snow cover changes by remote sensing should be added, and relevant comparison and discussion should be increased.
RESPONSE:Added to Introduction: “An alternative to field surveys is the use of optical satellite imagery, which has a long history of use in snow monitoring [31,32], with the MODIS[33-37] and Landsat[38-43] satellites frequently used. This subject has recently been reviewed [44].”
and:
“Further afield, Landsat tracked changes in snow cover extent from 1986-2018 in the South American Andes [45].”
Added references are: Meier, M. F. (1975). Application of remote-sensing techniques to the study of seasonal snow cover. Journal of Glaciology, 15(73), 251-265.
Schneider, S., & Matson, M. (1977). Satellite observations of snowcover in the Sierra Nevadas during the great California drought. Remote Sensing of Environment, 6(4), 327-334.
Lopez, P., Sirguey, P., Arnaud, Y., Pouyaud, B., & Chevallier, P. (2008). Snow cover monitoring in the Northern Patagonia Icefield using MODIS satellite images (2000–2006). Global and Planetary Change, 61(3-4), 103-116.
Marchane, A., Jarlan, L., Hanich, L., Boudhar, A., Gascoin, S., Tavernier, A., ... & Berjamy, B. (2015). Assessment of daily MODIS snow cover products to monitor snow cover dynamics over the Moroccan Atlas mountain range. Remote Sensing of Environment, 160, 72-86.
Deng, G., Tang, Z., Hu, G., Wang, J., Sang, G., & Li, J. (2021). Spatiotemporal dynamics of snowline altitude and their responses to climate change in the Tienshan Mountains, Central Asia, During 2001–2019. Sustainability, 13(7), 3992.
Tang, Z., Wang, X., Deng, G., Wang, X., Jiang, Z., & Sang, G. (2020). Spatiotemporal variation of snowline altitude at the end of melting season across High Mountain Asia, using MODIS snow cover product. Advances in Space Research, 66(11), 2629-2645.
Gunnarsson, A., Garðarsson, S. M., & Sveinsson, Ó. G. (2019). Icelandic snow cover characteristics derived from a gap-filled MODIS daily snow cover product. Hydrology and Earth System Sciences, 23(7), 3021-3036.
Park, S. H., Lee, M. J., & Jung, H. S. (2016). Spatiotemporal analysis of snow cover variations at Mt. Kilimanjaro using multi-temporal Landsat images during 27 years. Journal of Atmospheric and Solar-Terrestrial Physics, 143, 37-46.
McFadden, E. M., Ramage, J., & Rodbell, D. T. (2011). Landsat TM and ETM+ derived snowline altitudes in the Cordillera Huayhuash and Cordillera Raura, Peru, 1986–2005. The Cryosphere, 5(2), 419-430.
Cordero, R. R., Asencio, V., Feron, S., Damiani, A., Llanillo, P. J., Sepulveda, E., ... & Casassa, G. (2019). Dry-season snow cover losses in the Andes (18–40 S) driven by changes in large-scale climate modes. Scientific Reports, 9(1), 16945.
We expand the discussion by increasing comparisons to previous literature:
“A MODIS satellite study of year-round snow cover extent in Iceland [37] from 2000-2018 found a significant increase in snow cover for the month of June. A field study in the Australian Snowy Mountains [14] claimed a significant decline in summer snow patches. Both these study sites have maritime climates like that experienced in Scotland, with the Snowy Mountains being particularly similar as only a few small snow patches, if any, survive a typical austral summer. A Landsat study in the South American Andes [45] found significant declines in austral summer snow cover extent over the study period of 1986 to 2018. There have been numerous MODIS satellite studies in the High Mountain Area of Asia, these showing strong spatial and temporal heterogeneity in changes in snow cover extent [70-73].“
Added references:
Ackroyd, C., Skiles, S. M., Rittger, K., & Meyer, J. (2021). Trends in snow cover duration across river basins in high mountain Asia from daily gap-filled MODIS fractional snow covered area. Frontiers in Earth Science, 9, 713145.
Tang, Z., Deng, G., Hu, G., Zhang, H., Pan, H., & Sang, G. (2022). Satellite observed spatiotemporal variability of snow cover and snow phenology over high mountain Asia from 2002 to 2021. Journal of Hydrology, 613, 128438.
Tang, Z., Wang, X., Wang, J., Wang, X., Li, H., & Jiang, Z. (2017). Spatiotemporal variation of snow cover in Tianshan Mountains, Central Asia, based on cloud-free MODIS fractional snow cover product, 2001–2015. Remote Sensing, 9(10), 1045.
Chen, W., Ding, J., Wang, J., Zhang, J., & Zhang, Z. (2020). Temporal and spatial variability in snow cover over the Xinjiang Uygur Autonomous Region, China, from 2001 to 2015. PeerJ, 8, e8861.
4. Figure 4 is an important result supporting the conclusion of this paper, but it is insufficient to reflect the characteristics of the spatial distribution of remote sensing data in this paper with only a linear wave chart of interannual changes. It is suggested to increase the spatial expression.
RESPONSE: Due to the cloudiness of the Scottish Highlands it was not feasible to carry out a trend analysis for each of the 9 study sites individually. To improve our spatial analysis, however, we divide our study sites into an eastern and western subset, and add a trend analysis for these subsections. We add the following to Materials and Methods (section 2.3.3):
“Trend significance and magnitude were computed for 1) all nine study sites together; 2) an eastern subset (Cairngorms and Lochnagar) and 3) a western subset of study sites (Black Mount, Ben Nevis, |Glen Affric, Sgurr na Lapaich, Creag Meagaidh, Ben Alder and the Fannichs.) Any subsequent references to SV are for Case 1) all study sites, unless specifically noted otherwise. “
We add to the Results (section 3.3.1):
“For the eastern set of study sites (Cairngorms and Lochnagar) there were 103 Landsat images. The Mann-Kendall test still found a significant decline in SV from 1984 to 2022 (z=-2.24, p=0.025). For the western subset of study sites (Black Mount, Ben Nevis, Glen Affric, Sgurr na Lapaich, Creag Meagaidh, Ben Alder and the Fannichs) there were 189 Landsat images covering the study period, from which a significant decline in SV was also found (z=-2.18, p=0.03).
We add to the Study site section some additional information to inform our division into eastern and western subsections: “The Köppen-Geiger climate classification for the study sites is largely KT (polar, tundra), with some small areas of lower elevation classified as Dfc (cold, no dry season, cold summer.) [49] ...Due to the prevailing westerly winds, the western, Atlantic Ocean adjacent study sites have a higher mean annual precipitation than the eastern study sites. The eastern most two study sites of Cairngorms and Lochnagar both had an annual mean precipitation from 1991-2020 of less than 1900mm/year (mean for both study sites 1864mm/year), whilst the six more westerly study sites all had annual precipitations of greater than 2500 mm/year (mean for these western study sites of 3040mm/year). Temperatures were slightly higher for the western than the eastern subset, with mean annual temperatures of 3.44 and 3.78°C respectively. “
Added reference:
Beck, H. E., Zimmermann, N. E., McVicar, T. R., Vergopolan, N., Berg, A., & Wood, E. F. (2018). Present and future Köppen-Geiger climate classification maps at 1-km resolution. Scientific data, 5(1), 1-12.
Suggested references:
[1] Satellite observed spatiotemporal variability of snow cover and snow phenology over High Mountain Asia from 2002 to 2021. Journal of Hydrology,2022, 613, 128438.
[2] Spatiotemporal dynamics of snowline altitude and their responses to climate change in the Tienshan Mountains, Central Asia, During 2001–2019 [J]. Sustainability, 2021, 13(7): 3992.
[3] Spatiotemporal variation of snowline altitude at the end of melting season across High Mountain Asia, using MODIS snow cover product [J]. Advances in Space Research, 2020, 66(11): 2629-2645.
[4] Spatiotemporal variation of snow cover in Tianshan Mountains, Central Asia, based on cloud-free MODIS fractional snow cover product, 2001–2015 [J]. Remote Sensing, 2017, 9(10): 1045.
RESPONSE: All these references added to paper, either in the Discussion or Introduction.
Round 2
Reviewer 1 Report
The manuscript has been improved.
Figure 3 can be improved. It looks slightly crude.
Author Response
Reviewer comments in bold text, our responses in plain text.
The manuscript has been improved.
Figure 3 can be improved. It looks slightly crude.
RESPONSE: We've tried to improve Fig. 3: added Equation numbers to the processing steps, added shade to the data shapes, improved the arrows and tried to space it out so it looks less bunched together. We hope this is an improvement.
We want to thank the reviewer for their careful reviews of our paper. We know how time-consuming it can be. We feel your comments, especially as regards further explaining our method in the Materials and Method section, have strengthened our paper.
Reviewer 2 Report
I am satisfied with changes made by the authors and the their response.
I would suggest improving figure 1. Authors should add outline to Landcover maps. Legends in Elevation maps can be shifted to lower-right bottom side of the figure, utilising the space.
Author Response
Reviewer in bold text, our responses in plain text.
I am satisfied with changes made by the authors and the their response.
I would suggest improving figure 1. Authors should add outline to Landcover maps. Legends in Elevation maps can be shifted to lower-right bottom side of the figure, utilising the space.
RESPONSE: In Figure 1, we've shifted both the legends to blank space on the lower right-hand side to free up the main map. Outlines added to legend boxes.
We want to thank the reviewer for taking the time to look over this paper. We really appreciate the time and effort that it takes.
Reviewer 4 Report
-
The author made a good improvement or reply.
Author Response
The author made a good improvement or reply.
Thankyou for reviewing the paper. We know how much time it takes. We feel that your suggested changes, particularly as improving the analysis as regards spatial distribution, have improved the paper.